# Experimental Study on Dynamic Compression Mechanical Properties of Aluminum Honeycomb Structures

**Sheng Zhang [1], Wei Chen [1,*] , Deping Gao [1], Liping Xiao [2] and Longbao Han [3]**

[1] Jiangsu Province Key Laboratory of Aerospace Power System, College of Energy and Power Engineering, Nanjing University of Aeronautics and Astronautics, Nanjing 210016, China; zsnuaa@nuaa.edu.cn (S.Z.); gdp202@nuaa.edu.cn (D.G.)

[2] UAV Research Institute, Nanjing University of Aeronautics and Astronautics, Nanjing 210016, China; xiaolp@nuaa.edu.cn

[3] China International Engineering Consulting Corporation, Beijing 100048, China; hanlongbaonuaa@163.com

[*] Correspondence: chenwei@nuaa.edu.cn; Tel.: +86-025-8489-2220



**Featured Application: The SHPB test method with special measures is developed to study the dynamic compression mechanical properties of the aluminum honeycomb structures at a high strain rate. The stress hardening and softening effects are found, respectively.**

**Abstract:** In this paper, dynamic compression tests are developed to investigate the dynamic compression mechanical properties of the aluminum honeycomb structures at different strain rates, especially at the high strain rates. The difficulties at the high strain rates exist due to the large deformation, the low wave resistance and the size effect of the honeycomb structures. The Split Hopkinson Pressure Bar (SPHB) test method is carried out and special measures such as the adoption of waveform shaper, the size optimization of the impact bar and the specimen, and employment of the semiconductor strain gauge, etc. are taken to overcome the difficulties. It is discovered that the dynamic compression mechanical properties possess a stress hardening effect at a high strain rate from $1.3 \times 10^3$ s$^{-1}$ to $2.0 \times 10^3$ s$^{-1}$, but then a stress softening effect at a high strain rate of $4.6 \times 10^3$ s$^{-1}$. It is also discovered that the yield strength and the average plateau stress at the strain rate of $2.0 \times 10^3$ s$^{-1}$ is higher than that at the strain rate of $1.3 \times 10^3$ s$^{-1}$. However, the yield strength and the average plateau stress at the strain rate of $4.6 \times 10^3$ s$^{-1}$ is lower than that at the strain rate of $2.0 \times 10^3$ s$^{-1}$ and $1.3 \times 10^3$ s$^{-1}$, but higher than that at a quasi-static state. This indicates that the aluminum honeycomb structure is sensitive to the strain rate. Additionally, the damage mode of the aluminum honeycomb structure is plastic buckling, collapse and folding of the cell wall, which is carried out using dynamic compression tests. The folding length of the cell wall at a higher strain rate is found to be longer than that at a lower strain rate. The test results can also be used as the stress–strain curves of the honeycomb constitutive model at the high strain rates to carry out the numerical simulation of high-speed impact.

**Keywords:** aluminum honeycomb structure; dynamic compression mechanical properties; high strain rate; damage mode

## 1. Introduction

With advantages of low weight, large specific stiffness, and high specific strength, the metal honeycomb sandwich structure is extensively applied in the field of aerospace engineering. For example, the high-bypass turbofan engine casing is made from the honeycomb sandwich structure in order to

improve the impact resistance since the impact on the casing is a typical nonlinear physical process with the high strain rate up to $10^4$ s$^{-1}$ during the fan blade out events. The high strain rate has a great influence on the mechanical properties of the structure. The different honeycomb structures also possess different cell structure characteristics and mechanical properties, therefore, it is crucial to understand the dynamic compression mechanical properties of the aluminum honeycomb structures at a high strain rate.

The dynamic compression test is an effective way to obtain the dynamic mechanical properties of honeycomb materials. Equipment of dynamic compression tests mainly include dynamic material testing machine, the drop hammer impact table, Hopkinson bar, Taylor impact table and the high-speed impact test system, etc. Hopkinson bar is widely used at the medium and high strain rates due to the fewer restrictions. As early as 1914, Hopkinson [1] developed the Hopkinson compression bar test device. Kolsky [2] proposed a separated Hopkinson pressure bar setup on the basis of Hopkinson compression bar. Wu and Jiang [3] studied out-of-plane crushing properties of the honeycomb structures with a similar experimental device and obtained an increase of 74% of the crush strength under dynamical conditions compared to those under the quasi-static condition. The Split Hopkinson Pressure Bar (SHPB) method is widely used in dynamic compression tests of metal materials to obtain the dynamic mechanical properties. However, there are still many difficulties when it is applied to the honeycomb structure, especially at high strain rates due to several reasons. For example, the honeycomb specimen possesses the size effect [4] because of the diameter limit. The wave impedance of the honeycomb structure is lower than that of a conventional material bar which makes it difficult to form a strong signal in the transmission bar. On the other hand, the deformation of the honeycomb structure is large, and the nonuniform stresses may occur inside the specimen and undergo the multiple loading-unloading process [5]. The assumption of stress uniformity in the SHPB test for the honeycomb structure also needs to be further verified.

Due to the difficulties mentioned above, the SHPB test method cannot always provide satisfactory precision of the honeycomb structures under the impact loading. In recent years, viscoelastic bar has been used as a solution in the SHPB test method [6]. Zhao et al. [7,8] conducted viscoelastic bars and a two-strain measurement method on the SHPB test to improve the signal/noise ratio and to host larger samples containing a sufficient number of cells. They investigated the out-plane impact dynamic response of aluminum honeycomb materials at the medium (600 s$^{-1}$) and the low strain rates ($10^{-4}$ s$^{-1}$). The results indicated that the enhancement of the crushing strength occurred at the medium strain rate. Elnasri et al. [9] adopted the Nylon Hopkinson bar with a large diameter to investigate the existence of a shock front of the honeycomb and the Cymat foam at the low impact loading. It was found that no significant shock enhancement was observed for aluminum honeycomb structures and the sensitivity of the corresponding rate was not responsible for the strength enhancement.

On the basis of viscoelastic bar, the combined shear-compression impact test was presented [10]. Tounsi et al. [11] and Hou et al. [12] introduced the large diameter Nylon Split Hopkinson Pressure Bar system (NSHPB) with beveled ends of different angles to study the behavior of honeycomb structures under the multiaxial impact loadings. It was found that the impact strength of the honeycomb structure decreases with the increasing loading angle, while the shear strength changes in the opposite direction. In addition, Xu et al. [13,14] conducted an out-plane compression test of aluminum honeycomb structures with various dimensions, relative density, and honeycomb cell sizes, at the strain rate range of $5 \times 10^{-5}$ s$^{-1}$–$2 \times 10^2$ s$^{-1}$. The results indicate that the plateau stress generally increases with the strain rate and cellular structure of the honeycomb also affected the order of magnitude for the rate correlation. Wang et al. [15] adopted the dynamic impact test of Hopkinson bar to analyze the correlation between the strength and energy absorption capacity of the aluminum honeycomb materials under dynamic impact conditions. Generally speaking, previous studies have mainly focused on the improvements of traditional measurement. For example, viscoelastic bars with large diameters for SHPB are adopted to investigate the dynamic mechanical properties of honeycomb structure at the medium and the low

strain rates ($0$–$10^3$ s$^{-1}$). However, when a viscoelastic bar is applied to improve the measurement accuracy, there are more difficulties to obtain the stress, strain and strain rate accurately [16].

In this paper, the SHPB test method with several special measures such as optimization of the impact bar and the size of the specimen, adding the waveform shaper, and applying the semiconductor strain gauge firstly, etc. are explored to acquire the dynamic mechanical properties of the honeycomb structure at high strain rates. They are verified by two impact tests both on the impacted end and the supported end. The dynamic mechanical properties of the hexagonal aluminum honeycomb structures at high strain rates (higher than $10^3$ s$^{-1}$) are revealed for the first time. Afterwards, the compression damage mode of the aluminum honeycomb structure at high strain rates is studied.

## 2. Dynamic Compression Test at the High Strain Rate

### 2.1. Test Specimen

The regular hexagon aluminum honeycomb structure is adopted. The hexagonal aluminum honeycomb structure is commonly made of 0.02–0.1 mm thick aluminum foil through bonding. There are two manufacturing methods, forming and stretching. Stretch method is suitable for industrial production due to the high efficiency, so it is widely used. The process flow of the stretch method to manufacture aluminum honeycomb structure include aluminum foil cleaning, node glue, solidification, slitting, stretch. The cleaning process mainly contain alkali wash, rinsing, phosphoric anodization, spray and drying. Then J-70 adhesive based on the epoxy resin is applied, which can inhibit corrosion. The gluing process is completed by a special gluing machine. After that, the coated aluminum foil needs to be folded to form a panel and then curing. The curing parameters are related to the selected adhesive, generally speaking, the pressure is 0.5 Mpa, the time is 3–5 min. After slitting and stretching forming, the honeycomb panel is formed.

The basis material of the aluminum honeycomb structure is 5052 aluminum alloy. The chemical composition is shown in Table 1. The cell characteristics are as follows: the cell wall thickness $t$ is 0.06 mm with the length $l$ of 1.732 mm; cell structure characteristic size $d$ is 3 mm, as shown in Figure 1. The wall thickness of the opposite cells in each cell structure is 0.12 mm due to the manufacturing method. The thickness of the aluminum honeycomb structure sample is 5 mm, and 3 samples were prepared for each test. The coordinate system of the aluminum honeycomb structure is shown in Figure 2. The direction $x_3$ is the direction of the out-of-plane loading, and $x_1$ and $x_2$ are the directions of the in-plane loading.

**Table 1.** The chemical composition of 5052 aluminum alloy.

| Alloying Element | Fe | Cu | Mn | Mg | Cr | Zn | Si | Impurity |
|---|---|---|---|---|---|---|---|---|
| Percentage (%) | ≤0.40 | ≤0.10 | ≤0.10 | 2.2~2.8 | 0.15~0.35 | ≤0.10 | ≤0.25 | ≤0.15 |

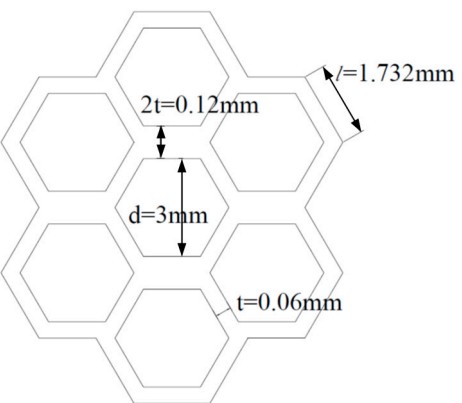

**Figure 1.** Cell characteristics of the aluminum honeycomb.

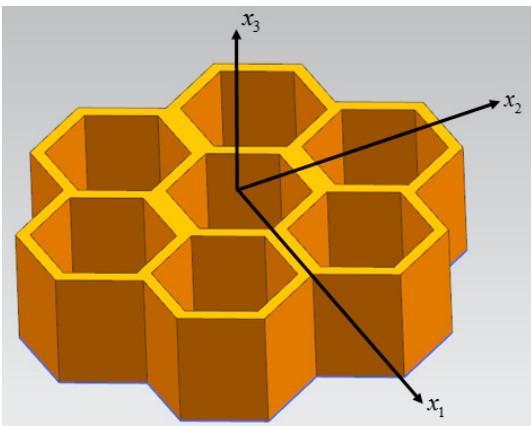

**Figure 2.** Coordinate system of aluminum honeycomb structure.

### 2.2. The Special Measures of Split Hopkinson Pressure Bar (SHPB) Test Method and Verification

The SHPB method is widely used in dynamic compression tests for metal materials to obtain the dynamic mechanical properties. However, there are still many difficulties when it is applied to the honeycomb structure, especially at the high strain rate, which are mentioned before. The following special measures are taken to overcome the difficulties.

The actual impact area of the honeycomb structure is small compared to metallic materials due to the thin cell wall during the impact, which may cause the stress nonuniformity of the specimen and the divergence of the stress wave. The specimen of Φ32 is applied to reduce the stress nonuniformity and the divergence of the stress wave as well as the diameter of the incident bar is 37 mm. The specimen contains nine complete cells in the direction of diameter. Try to maintain the integrity of cells at the edge of the specimen during the processing.

The wave impedance of the honeycomb structure is lower than that of aluminum alloy bar which makes the weak signal in the transmission bar. The semiconductor strain gauge is applied on the transmission bar to acquire the nice waveform of the weak signal, which cannot be measured by the conventional resistance strain gauge. It is also more convenient than the adoption of the viscoelastic bar. The contact surface between the bar and the specimen is lubricated to reduce the friction between the end face of the specimen, the incident bar as well as the transmission bar.

The loads between the impact end and the support end of the specimen may be different due to the divergent oscillation of incident wave, resulting in poor stress uniformity inside the specimen. Therefore, the brass wafer is pasted at the impacted end of the incident bar to eliminate the unbalanced stress caused by the divergence of the stress wave.

The short specimen and long impact bar are employed to make the stress wave passing through at least 3–4 reflections in the specimen, realizing the homogenization of the stress in the specimen.

The measurements based on the SHPB method are shown in Figure 3.

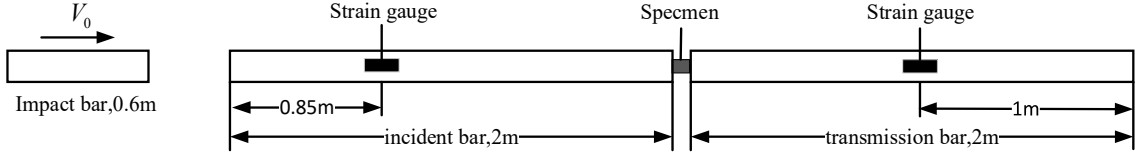

**Figure 3.** Split Hopkinson Pressure Bar (SHPB) test system with measurements.

The length of the impact bar is 0.6 m, and those of the incident and transmission bars are 2 m. The strain gauges are pasted at 0.85 m from the impact end of the incident bar and the middle of transmission bar respectively, to obtain the strain curves of the incident and transmission bars in the

loading process, and then the stress–strain relationship of the specimens can be obtained. The size of the specimen of Φ32 × 5 mm is adopted in the dynamic compression tests.

The measurements based on the SHPB method are verified by the direct impact tests of impact end and support end which are modified from the SHPB test. The stress responses of the impact end and the support end of the aluminum honeycomb structure are obtained, respectively, by the twice impact method, so as to verify the effectiveness of the measurements of SHPB test method.

The direct impact tests of the support end are shown in Figure 4. The aluminum honeycomb structure specimen is contacted with the transmission bar in the SHPB test system, and the semiconductor strain gauge is pasted on the transmission bar. The distance between strain gauge and the impact end of the bar is 0.85 m. The incident bar is removed, and a long elastic impact bar is adopted to impact the specimen at a certain velocity directly. It is called test 1.

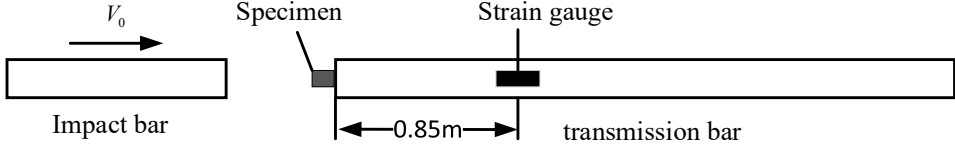

**Figure 4.** Stress test scheme of specimen support end (test 1).

The direct impact tests of the impact end are shown in Figure 5. The specimen is contacted with the end of the impact bar, then the specimen with the impact bar hit the transmission bar at a certain velocity. The strain is measured by the semiconductor strain gauge on the transmission bar, and then we can obtain the stress response through calculation. It is called test 2.

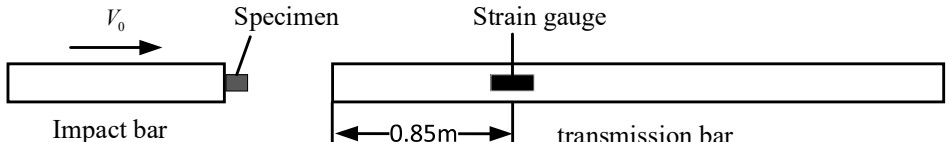

**Figure 5.** Stress test scheme of specimen impact end (test 2).

The pictures of the direct impact tests are shown in Figures 6 and 7 respectively.

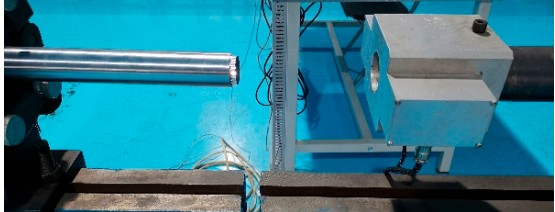

**Figure 6.** The picture of support end test.

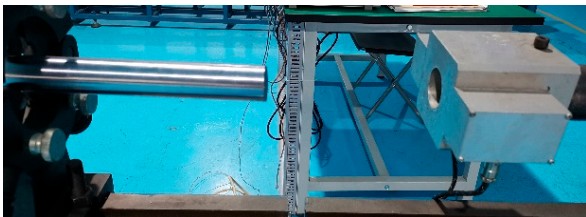

**Figure 7.** The picture of impact end test.

The strain-time curve of the transmission bar is obtained by the strain gauge measurement in the direct impact tests of both impact end and support end. The transmission bar is made of hard

aluminum alloy, which causes elastic deformation during the impact. The stress of the transmission bar can be calculated as

$$\sigma_{bar}(t) = E_{bar}\varepsilon_{bar}(t) \tag{1}$$

The force on the contact surface between the honeycomb structure and the transmission bar can be calculated as

$$F = s_{bar}E_{bar}\varepsilon_{bar}(t) \tag{2}$$

The stress of the honeycomb structure can be calculated as

$$\sigma_h(t) = \frac{s_{bar}}{s_h}E_{bar}\varepsilon_{bar}(t) \tag{3}$$

where $\sigma_h(t)$ is the stress of the honeycomb structure, $s_{bar}$ is the sectional area of the transmission bar, $s_h$ is the initial macroscopic sectional area of the honeycomb structure, $E_{bar}$ is the elasticity modulus of the transmission bar, and $\varepsilon_{bar}(t)$ is the strain-time curve of the transmission bar. It is noticed that the sectional area of the honeycomb structure $s_h$ varies during the impact, so the stress of the honeycomb structure $\sigma_h(t)$ is the nominal stress.

The SHPB test with measurements is conducted in order to compare the stress response of the direct impact test and SHPB test. It's called test 3.

The impact velocity at high strain rate is adopted in the tests. The impact velocities of three tests are 13.83 m/s, 14.41 m/s and 13.85 m/s respectively. The results of the tests are shown in Table 2, and the stress curve is shown in Figure 8.

**Table 2.** Impact test results of three tests.

| Parameter | Test 1 | Test 2 | Test 3 |
| --- | --- | --- | --- |
| Impact velocity (m/s) | 13.83 | 14.41 | 13.85 |
| Yield strength (MPa) | 2.62 | 2.73 | 2.61 |
| Yield time (ms) | 0.046 | 0.026 | 0.048 |
| Mean nominal plateau stress (MPa) | 1.616 | 1.682 | 1.615 |

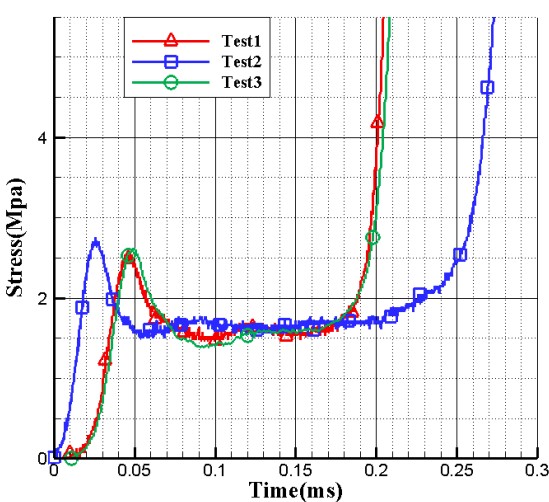

**Figure 8.** Stress–time curve of direct impact tests and SHPB test.

There is a time delay about the stresses compared to each test, as shown in Figure 8. The time delay is caused by the distance between the impact end and the support end of the specimen. The time delay between the impact end and the support end is 0.02 ms.

There is not only a time delay between each test, but also a stress value difference. The equilibrium stress factor $R(\sigma)$ of the specimen is defined as the ratio of the difference of the maximum and minimum

mean nominal plateau stress to the average value. The equilibrium stress factor can measure the stress uniformity. The calculation formula is as follows

$$R(\sigma) = \frac{\overline{\sigma}_{\text{max}} - \overline{\sigma}_{\text{min}}}{\overline{\sigma}} \times 100\% \tag{4}$$

where $\overline{\sigma}_{\text{max}}$ is the maximum mean nominal plateau stress of the three tests, $\overline{\sigma}_{\text{min}}$ is the minimum mean nominal plateau stress of the three tests, $\overline{\sigma}$ is the average value of $\overline{\sigma}_{\text{max}}$ and $\overline{\sigma}_{\text{min}}$. In general, when $R(\sigma)$ < 5%, it is considered the stress is uniform [17]. The test results show that the stress factor $R(\sigma)$ of three tests is 4.06%, which meets the requirement of stress uniformity in the specimen. It is verified that the special measures of SHPB method can be used in the dynamic compression test of the aluminum honeycomb structures at the high strain rate.

The dynamic compression tests of aluminum honeycomb structures at a strain rate of $1 \times 10^3$ s$^{-1}$, $2 \times 10^3$ s$^{-1}$ and $5 \times 10^3$ s$^{-1}$ are carried out by adjusting the impact velocity of the impact bar to obtain the different strain rates. Three repetitions are performed at each strain rate.

### 2.3. Test Results and Analysis

#### 2.3.1. Test Results at Strain Rate $1 \times 10^3$ s$^{-1}$

The results of the dynamic compression tests under the strain rate at $1 \times 10^3$ s$^{-1}$ are shown in Table 3. The average strain rate at the stable stage is approximately $1.3 \times 10^3$ s$^{-1}$. The strain rate curves and the stress–strain curves are shown in Figures 9 and 10 respectively. The test results show that the maximum strain is only 32% at the strain rate $1.3 \times 10^3$ s$^{-1}$, and the average compression rate is 93.78%. After the elastic stage, the specimen enters into the plateau stage and appears in the stress oscillation once. However, the specimen does not reach the densification.

**Table 3.** The results of the dynamic compression tests under the strain rate of $1 \times 10^3$ s$^{-1}$.

| Test No | Impact Velocity (m/s) | Thickness of Specimen before Test (mm) | Thickness of Specimen After Test (mm) | Compression Ratio (%) | Strain Rate Average at Plateau Stage (s$^{-1}$) | Initial Collapse Stress (MPa) | Initial Collapse Strain |
|---|---|---|---|---|---|---|---|
| 1 | 4.63 | 4.99 | 0.31 | 93.76 | $1.26 \times 10^3$ | 2.87 | 0.057 |
| 2 | 4.81 | 4.97 | 0.30 | 93.96 | $1.34 \times 10^3$ | 2.91 | 0.059 |
| 3 | 4.67 | 5.02 | 0.32 | 93.62 | $1.29 \times 10^3$ | 2.89 | 0.060 |

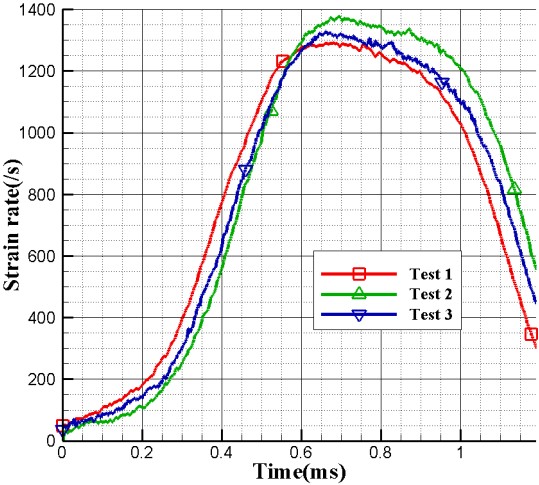

**Figure 9.** Strain rate curve at strain rate $1.3 \times 10^3$ s$^{-1}$.

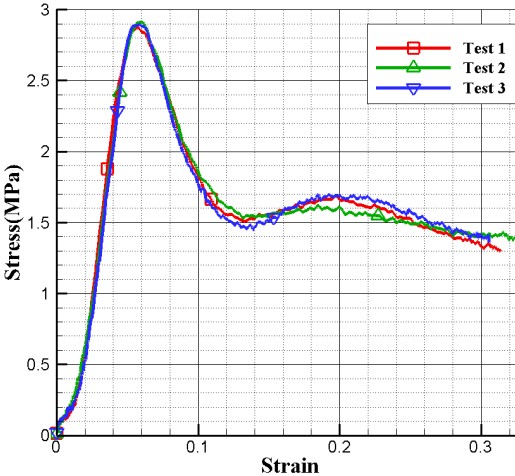

**Figure 10.** Stress–strain curve at strain rate $1.3 \times 10^3$ s$^{-1}$.

To obtain the macroscopic mechanical property of aluminum honeycomb structures at a strain rate of $1.3 \times 10^3$ s$^{-1}$, the average value of the strain rate curve and stress–strain curve from the three tests are obtained after interpolation.

### 2.3.2. Test Results at Strain Rate $2 \times 10^3$ s$^{-1}$

The results of the dynamic compression tests under the strain rate of $2 \times 10^3$ s$^{-1}$ are shown in Table 4. The average strain rate at the stable stage is approximately $2 \times 10^3$ s$^{-1}$. The strain rate curves are shown in Figure 11, and the stress–strain curves are shown in Figure 12. The test results show that the maximum strain can reach 48% at the strain rate of $2 \times 10^3$ s$^{-1}$. The average compression ratio of specimens reaches 94.73%, and it is 0.95% higher than that at the strain rate of $1.3 \times 10^3$ s$^{-1}$. Due to the strain rate increasing, the specimen enters into the plateau stage after the elastic stage, and appears in the stress oscillation twice, however, the specimen does not still reach the densification.

**Table 4.** The results of the dynamic compression tests under the strain rate $2 \times 10^3$ s$^{-1}$.

| Test No | Impact Velocity(m/s) | Thickness of Specimen before Test (mm) | Thickness of Specimen after Test (mm) | Compression Ratio (%) | Strain Rate Average at Plateau Stage (s$^{-1}$) | Initial Collapse Stress (MPa) | Initial Collapse Strain |
|---|---|---|---|---|---|---|---|
| 1 | 6.62 | 5.01 | 0.28 | 94.41 | $1.96 \times 10^3$ | 2.96 | 0.047 |
| 2 | 6.64 | 4.97 | 0.25 | 94.97 | $2.08 \times 10^3$ | 2.99 | 0.046 |
| 3 | 6.65 | 5.02 | 0.26 | 94.82 | $1.96 \times 10^3$ | 3.0 | 0.047 |

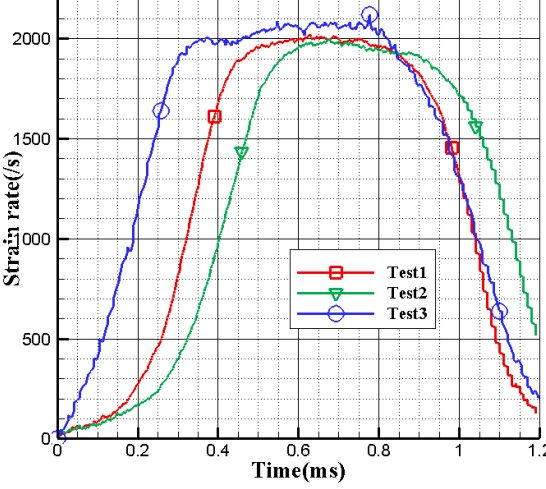

**Figure 11.** Strain rate curve at strain rate $2.0 \times 10^3$ s$^{-1}$.

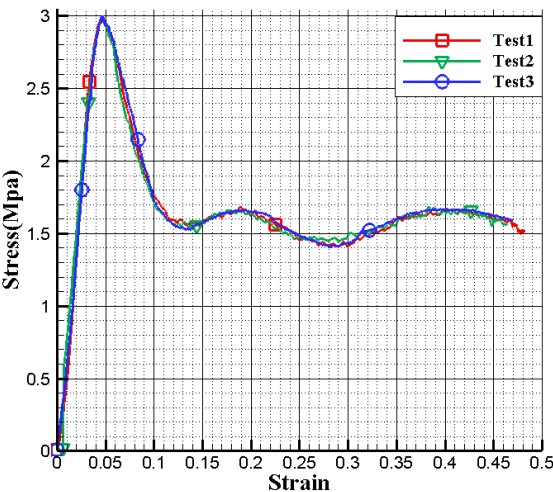

**Figure 12.** Stress–strain curve at strain rate $2.0 \times 10^3$ s$^{-1}$.

To obtain the macroscopic mechanical property of the aluminum honeycomb structure at strain rate of $2.0 \times 10^3$ s$^{-1}$, the average value of the strain rate curve and stress–strain curve from the three tests are obtained after interpolation.

### 2.3.3. Test Results at Strain Rate $5 \times 10^3$ s$^{-1}$

The results of the dynamic compression tests under the strain rate $5 \times 10^3$ s$^{-1}$ are shown in Table 5. The average strain rate at the stable stage is approximately $4.6 \times 10^3$ s$^{-1}$. The strain rate curves are shown in Figure 13, and the stress–strain curves are shown in Figure 14. The test results show that there is a slight oscillation in the stress obtained from the test, which is attributed to the weak divergence of the stress wave under the high strain rate. The time history of the strain rate and the stress–strain still present in a good repeatability.

**Table 5.** The results of the dynamic compression tests under the strain rate of $4.6 \times 10^3$ s$^{-1}$.

| Test No | Impact Velocity(m/s) | Thickness of Specimen before Test (mm) | Thickness of Specimen after Test (mm) | Compression Ratio (%) | Strain Rate Average at Plateau Stage (s$^{-1}$) | Initial Collapse Stress (MPa) | Initial Collapse Strain |
|---------|---------|---------|---------|---------|---------|---------|---------|
| 1 | 13.85 | 4.97 | 0.16 | 96.78 | $4.59 \times 10^3$ | 2.61 | 0.051 |
| 2 | 13.95 | 4.98 | 0.16 | 96.79 | $4.61 \times 10^3$ | 2.58 | 0.053 |
| 3 | 13.82 | 5.03 | 0.17 | 96.63 | $4.58 \times 10^3$ | 2.62 | 0.064 |

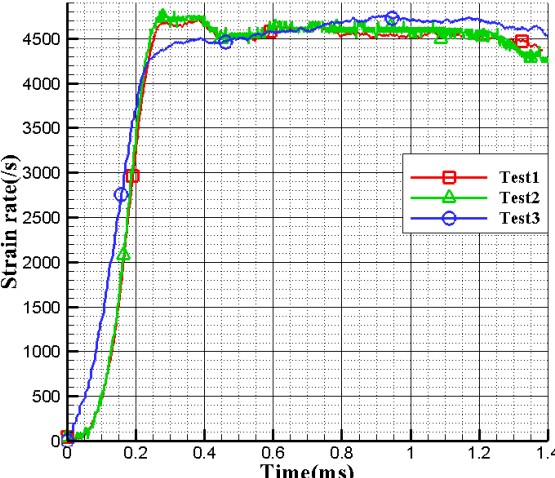

**Figure 13.** Strain rate curve at strain rate $4.6 \times 10^3$ s$^{-1}$.

It can be seen that the maximum strain reaches 79% at strain rate $4.6 \times 10^3$ s$^{-1}$, and the average compression rate of the specimen reaches 96.73%, which is 2.95% higher than that at the strain rate of $1.3 \times 10^3$ s$^{-1}$, and 2% higher than that of the strain rate of $2.0 \times 10^3$ s$^{-1}$. It can be seen that the impact velocity increases, the strain rate is significantly enhanced. The specimen reaches the densification after the plateau stage.

To obtain the macroscopic mechanical property of aluminum honeycomb structure at strain rate of $4.6 \times 10^3$ s$^{-1}$, the average value of the strain rate curve and stress–strain curve from the two tests are obtained after interpolation.

There are slight deviations of each test result at the same strain rate. The reason is because the thickness of aluminum foil varies during processing, and the boundary difference of the specimen when fabricated by wire-cutting.

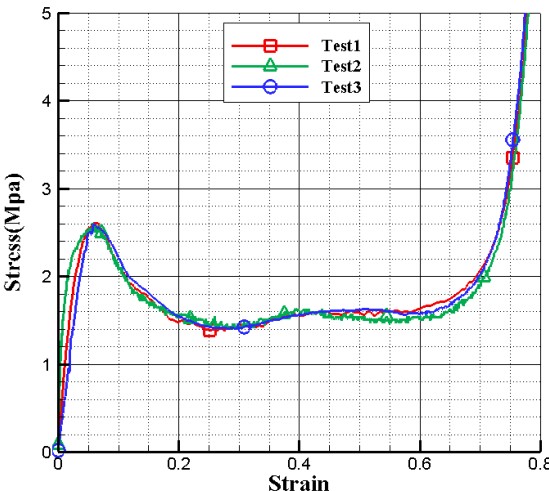

**Figure 14.** Stress–strain curve at strain rate $4.6 \times 10^3$ s$^{-1}$.

### 2.3.4. Test Results of Quasi-Static

The quasi-static compression tests are carried out by the electronic universal testing machine at room temperature via displacement loading method. The loading rate is 0.2 mm/min, and the sampling time interval is 0.2 s, the test will not stop until the specimen is obviously compacted. The three quasi-static compression tests are conducted by the universal electronic testing machine at the strain rate of $6.67 \times 10^{-4}$ s$^{-1}$. The average stress–strain curve of the aluminum honeycomb structure under quasi-static compression is obtained, as shown in Figure 15.

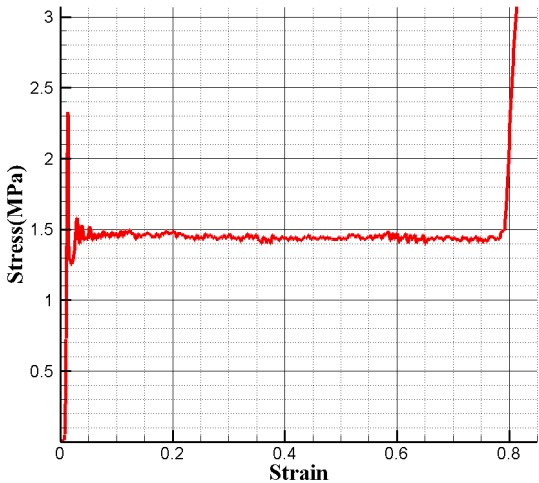

**Figure 15.** Stress–strain curve of Quasi-static compression test.

As is shown in Figure 15, the yield strength (initial collapse stress) $\sigma_{c0}$ = 2.33 MPa, and the corresponding initial collapse strain is $\varepsilon_{c0}$ = 1.33 × 10$^{-2}$. During the plateau stage, the strain increases from 0.188 to 0.792. The stress oscillates slightly, but it remains with the constant of 1.43 MPa. When the strain increases to the initial strain of densification at $\varepsilon_{d0}$ = 0.792, the slope of the stress–strain curve changes significantly, and the specimen is gradually compacted.

### 2.4. Comparison at Different Strain Rates

The mean values of the dynamic compression test results (the strain rate of 1.3 × 10$^3$ s$^{-1}$, 2.0 × 10$^3$ s$^{-1}$, 4.6 × 10$^3$ s$^{-1}$, 6.67 × 10$^{-4}$) are fitted into the strain rate curves and the stress–strain curves at different strain rates, as shown in Figures 16 and 17, respectively.

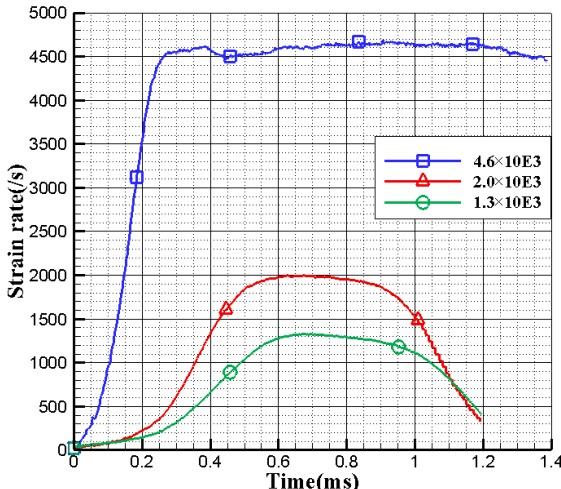

**Figure 16.** Mean stress rate curve at various strain rate.

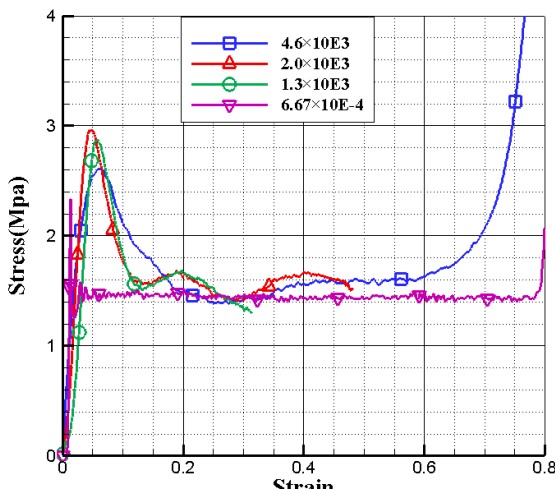

**Figure 17.** Mean stress–strain curve at various strain rate.

The compression of the quasi-static and the strain rate of 4.6 × 10$^3$ s$^{-1}$ both experience the elastic stage, the plateau stage and the densification stage. However, the strain rates of 1.3 × 10$^3$ s$^{-1}$ and 2.0 × 10$^3$ s$^{-1}$ only experience the elastic stage and the plateau stage, as shown in Figure 17. It can be seen that there is no fluctuation at Quasi-static state, only one fluctuation at strain rate of 1.3 × 10$^3$, two fluctuations at a strain rate of 2.0 × 10$^3$, and three fluctuations at a strain rate of 4.6 × 10$^3$. The fluctuation appears in the plateau stage, which is because the cellular wall instability and buckling during the compression. It can be seen that the fluctuations increase with the strain rate increases.

The yield strength, the average plateau stress and the initial densification strain of the aluminum honeycomb structure at different strain rates are shown in Table 6. The yield strength ratio at the strain rate of $2.0 \times 10^3$ $s^{-1}$ to the strain rate of $1.3 \times 10^3$ $s^{-1}$ is 1.03:1, while the average plateau stress ratio of them is 1.02:1. Additionally, the yield strength and the average plateau stress at the strain rate of $2.0 \times 10^3$ $s^{-1}$ and $1.3 \times 10^3$ $s^{-1}$ are both higher than that at the strain rate $6.67 \times 10^{-4}$. It is indicated that the aluminum honeycomb structure processes the stress hardening effect.

The yield strength ratios at the strain rate $4.6 \times 10^3$ $s^{-1}$ to the strain rate $1.3 \times 10^3$ $s^{-1}$ and $2.0 \times 10^3$ $s^{-1}$ are 0.91:1 and 0.88:1, respectively, while the average plateau stress ratios of them are 0.95:1 and 0.93:1. It is indicated that the dynamic compression mechanical properties at the high strain rate of $4.6 \times 10^3$ $s^{-1}$ process the stress softening effect.

**Table 6.** Yield strength, average plateau stress and initial densification strain at different strain rates.

| Strain rates ($s^{-1}$) | $6.67 \times 10^{-4}$ | $1.3 \times 10^3$ | $2.0 \times 10^3$ | $4.6 \times 10^3$ |
|---|---|---|---|---|
| Yield strength $\sigma_{c0}$ (MPa) | 2.33 | 2.88 | 2.97 | 2.61 |
| Average plateau stress (MPa) | 1.44 | 1.58 | 1.61 | 1.50 |
| Initial densification strain ($\varepsilon_{d0}$) | 0.792 | — | — | 0.68 |

The stress hardening effect and softening effect are both the combination of the characteristic of the honeycomb structure itself and the sealed gas effect. The characteristic of the honeycomb structure makes it exhibit higher stress in the dynamic compression tests at the high strain rate, however, the sealed gas effect plays the dominant role.

As the loading time is short in the dynamic compression tests at the high strain rate, the cell walls are not damaged, and the gas in the honeycomb cells can be approximately sealed, leading to the higher gas pressure. As the impact bar velocity increases, the yield strength and the plateau stress at the strain rate of $2.0 \times 10^3$ $s^{-1}$ are higher than those at the strain rate of $1.3 \times 10^3$ $s^{-1}$. However, as the impact bar velocity increases further, the cell walls seems partly damaged in the dynamic compression tests at the strain rate of $4.6 \times 10^3$ $s^{-1}$ and the gas in the honeycomb cells can be partly sealed, leading to the lower gas pressure, so the yield strength and the plateau stress at the strain rate of $4.6 \times 10^3$ $s^{-1}$ are lower than those at the strain rate of $1.3 \times 10^3$ $s^{-1}$ and $2.0 \times 10^3$ $s^{-1}$. Additionally, the gas flow out fully and the sealed gas effect is negligible in the process of the quasi-static compression, leading to the ambient gas pressure in the honeycomb cells, therefore the yield strength and the plateau stress at quasi-static are the lowest.

The initial densification strain is adopted as a material characteristic parameter to describe the crushability of the aluminum honeycomb structure in the maximum deformation of the plateau stage under the quasi-static and dynamic conditions. The dynamic compression at the strain rates of $1.3 \times 10^3$ $s^{-1}$ and $2.0 \times 10^3$ $s^{-1}$ does not reach the densification. The initial densification strain at the strain rate of $4.6 \times 10^3$ $s^{-1}$ is smaller than that of the quasi-static stage, it is shown that the strain of the maximum deformation at a high strain rate is smaller than that at the quasi-static state.

## 3. The Damage Mode

The scanning electron microscopy of the cell structure after impact is show in Figure 18. The damage mode of the honeycomb structure is observed during the test. The damage mode mainly includes cell wall plastic buckling, collapse, and compaction. The elastic buckling primarily emerges in the cell wall at the elastic stage. Both ends of the cell wall are neither completely free nor fixedly clamped, as shown in Figure 19. When loading to the yield strength, the plastic buckling occurs at the cell wall. The collapse mode is periodically gradual folding with the length of λ along the honeycomb thickness. The folding of cell walls is hardly compression, which is primarily achieved by bending. With the spread of the stress wave, the adjacent cell walls experience plastic buckling and collapse, and gradually spread out, leading to the compaction of the whole structure. The scanning electron microscopy of the cell wall folding is show in Figure 20.

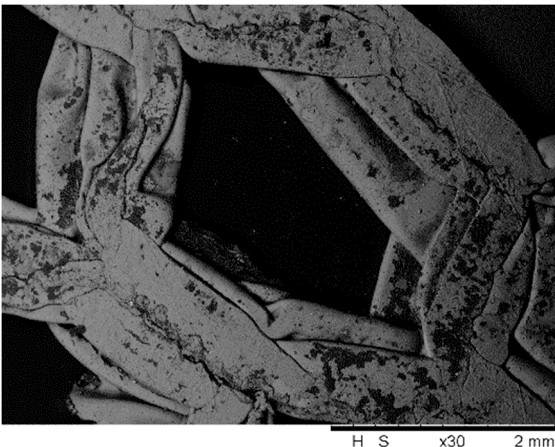

**Figure 18.** Electron micrograph of the cell structure after impact.

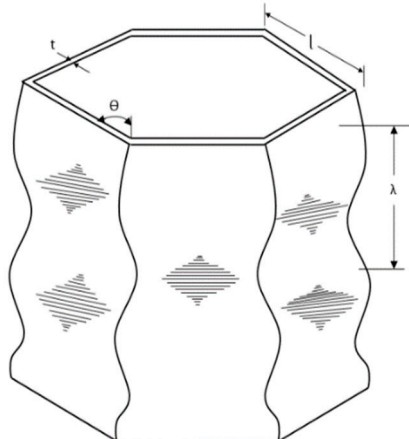

**Figure 19.** Elastic buckling of the hexagonal cell wall.

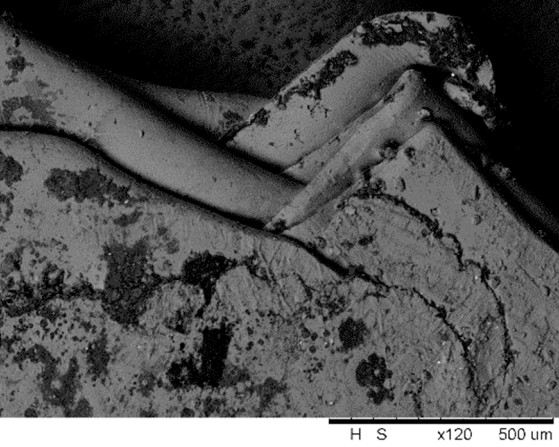

**Figure 20.** Electron micrograph of the cell wall folding.

The specimen after dynamic compression test of quasi-static, at the strain rates of $1.3 \times 10^3$ s$^{-1}$, $2.0 \times 10^3$ s$^{-1}$ and $4.6 \times 10^3$ s$^{-1}$ are shown in Figure 21. The average folding length of the specimen at each strain rate has been measured, as shown in Table 7. The specimen at high strain rate ($4.6 \times 10^3$ s$^{-1}$) has the longest folding length of 1.49 mm, and that at quasi-static has the shortest folding length of 0.59 mm, which also indicates the sensitivity of the strain rate. It can be seen that the folding length of the specimen increases as the strain rate increases.

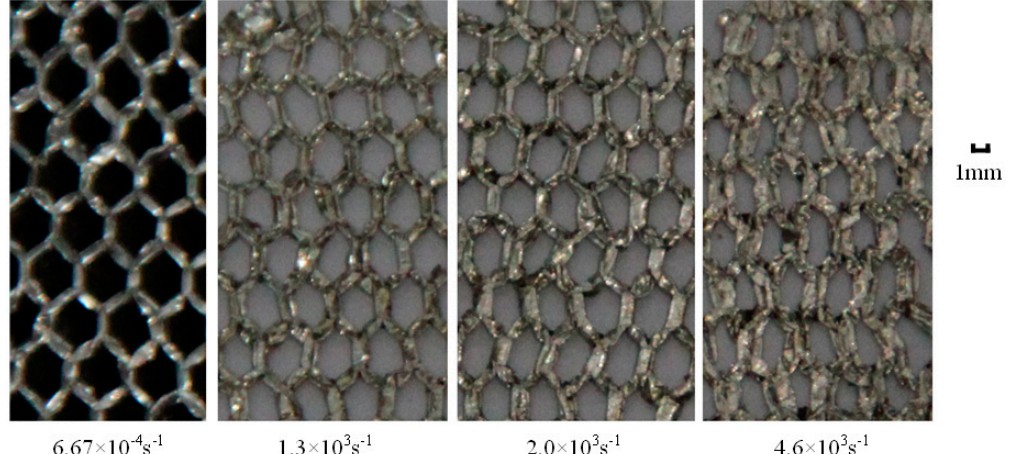

**Figure 21.** The specimen after dynamic compression test at various strain rates.

**Table 7.** The folding length of the specimen at various strain rates.

| Strain rate(s$^{-1}$) | $6.67 \times 10^{-4}$ | $1.3 \times 10^3$ | $2.0 \times 10^3$ | $4.6 \times 10^3$ |
|---|---|---|---|---|
| folding length(mm) | 0.59 | 0.89 | 0.96 | 1.49 |

## 4. Conclusions

The dynamic compression mechanical properties of aluminum honeycomb structures at high strain rates, together with a comparison among the quasi-static state are investigated in this paper by the dynamic compression tests of the aluminum honeycomb structures. The followings are discovered in the research:

(1) The stress hardening and softening effects of the aluminum honeycomb structure are both observed, which indicates that the aluminum honeycomb structure is sensitive to the strain rate.

(2) The damage modes of the aluminum honeycomb structures under dynamic compression include cell wall plastic buckling, collapse, and compaction. The folding length of the cell wall at a high strain rate is longer than that at a low strain rate, which also indicates the sensitivity of the strain rate.

(3) The SHPB test method with special measures can be used to acquire the dynamic mechanical properties of the metal honeycomb structures at high strain rates.

**Author Contributions:** Conceptualization, S.Z. and W.C.; methodology, S.Z.; validation, S.Z., and L.H.; formal analysis, S.Z., and L.H.; investigation, S.Z.; resources, W.C.; data curation, D.G.; writing—original draft preparation, S.Z.; writing—review and editing, W.C., D.G. and S.Z.; project administration, L.X.; funding acquisition, L.X. All authors have read and agreed to the published version of the manuscript.

**Funding:** This research was supported by the Fundamental Research Funds for the Central Universities, NO. NS2019060.

**Acknowledgments:** I would like to show my deepest gratitude to another supervisor, Chaoping Zang, a respectable, responsible and resourceful scholar. Without his enlightening instruction, impressive kindness and patience, I could not have completed my article.

**Conflicts of Interest:** The authors declare no conflict of interest.

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
