# Peer review of "Experimental Study on Dynamic Compression Mechanical Properties of Aluminum Honeycomb Structures"

_applsci, doi:10.3390/app10031188_

Round 1

Reviewer 1 Report

The study that the authors have conducted concerns the behavior of aluminium honeycomb structure to dynamic compression at different impact velocities. In general, the study conducted here seems to be very promising and important for the field of aerospace engineering. I recommend minor revision of manuscript. The suggestions are found in the attached file as sticky comments. I recommend you see another work similar to yours and presented more precisely from methodology point of view (Stanczak, M.; Fras, T.; Blanc, L.; Pawlowski, P.; Rusinek, A. Blast-Induced Compression of a Thin-Walled Aluminum Honeycomb Structure—Experiment and Modeling. Metals 20199, 1350).

Author Response

Please see the attachment, thanks.

Reviewer 2 Report

In this paper, the authors advanced the SHPB test method with several special measures to optimize the impact bar and size of the specimen, etc. By applying these improvements, they measured dynamic mechanical properties of honeycomb structure made of an Al-alloy at the high strain rates.

1) The article is not written with a sufficient care. In many cases, the indices are not in the proper positions, there is no spaces between values and units (e.g. "d0=3mm" instead of "d0 = 3 mm". There are also many grammatical flaws: Line 19 incorrect passive tense "are develop" instead of "are developed"; mixing singular and plural: Line 139 "The measurements ... is based...". Thus the article needs fine polishing.

2) There is no clear distinction between "experimental" and "result" section.

3) There are too many figures, and the figure captions are too scarce since they do not provide enough information.

Experimental part.

4) What was the chemical composition of the alloy?

5) How the cells were made?

6) Details about pressure brazing.

7) Which machine was used for quasi-static testing?

Results

8) There are no microstructures.

9) The images in Fig. 23 are not clear, and it is difficult to discern significant information from them. More clear explanation is needed, also some labels on the images would be welcome. There are no scale bars indicating the size in Fig. 23.

Discussion:

10) More discussion is needed for the explanation of the results. There is no comparison with other materials tested at high strain rates.

11) Fig. 21. Authors observed fluctuations on stress-strain curves. They require more explanations.

12) Table 5. Only average values are given. What about deviations?

Author Response

Comment: The article is not written with a sufficient care. In many cases, the indices are not in the proper positions, there is no spaces between values and units (e.g. "d0=3mm" instead of "d0= 3 mm". There are also many grammatical flaws: Line 19 incorrect passive tense "are develop" instead of "are developed"; mixing singular and plural: Line 139 "The measurements ... is based...". Thus the article needs fine polishing.

Response: We are sorry that there are some mistakes in the article. We checked the whole article carefully and corrected some errors.

Comment: There is no clear distinction between "experimental" and "result" section.

Response: Modifications and supplements are made in the "experimental" section. Description and analysis are added to the "result" section.

Comment: There are too many figures, and the figure captions are too scarce since they do not provide enough information.

Response: The mean strain rate curve and the mean stress-strain curve at each strain rate has been removed. The figure captions has been checked and modified.

Comment: What was the chemical composition of the alloy?

Response: The basis material of the aluminum honeycomb structure is 5052 aluminum alloy. The chemical composition is shown in Table 1. We have added these information in the revised manuscript.

Table 1 The chemical composition of 5052 aluminum alloy

Alloying

element

Fe

Cu

Mn

Mg

Cr

Zn

Si

Impurity

Percentage(%)

≤0.40

≤0.10

≤0.10

2.2~2.8

0.15~0.35

≤0.10

≤0.25

≤0.15

Comment: How the cells were made? Comment: Details about pressure brazing.

Response: The hexagonal aluminum honeycomb structure is commonly made of 0.02-0.1mm thick aluminum foil through bonding. There are two manufacturing methods, forming and stretching. stretching method is suitable for industrial production due to the high efficiency, so it is widely used. The process flow of the stretch method to manufacture aluminum honeycomb structure is Aluminum foil cleaning, Node glue, solidification, slitting, stretch.

Cleaning process mainly contain alkali wash, rinsing, phosphoric anodization, spray and drying. J-70 adhesive based on the epoxy resin is applied, which can inhibit corrosion. The gluing process is completed by a special gluing machine. The coated aluminum foil needs to be folded to form a panel and then curing. The curing parameters are related to the selected adhesive, generally speaking, the pressure is 0.5Mpa, the time is 3-5min. After slitting and stretching forming, the honeycomb panel is formed.

Comment: Which machine was used for quasi-static testing?

Response: The quasi-static compression tests are carried out by the electronic universal testing machine at room temperature via displacement loading method. The loading rate is 0.2mm/min, and the sampling time interval is 0.2s, the test will not stop until the specimen is obviously compacted. We have added these information in the revised manuscript.

Comment: There are no microstructures.

Response: The scanning electron microscopy of the cell after impact is show in Figure 2. The scanning electron microscopy of the cell wall folding form is show in Figure 3. It can be clearly seen that the folding form is interval cascade folding between adjacent cell walls. After that, “damage mode” section has been revised and supplemented.

Figure 2 scanning electron micrograph of the cell after impact

Figure 3 scanning electron micrograph of the cell wall folding form

Comment: The images in Fig. 23 are not clear, and it is difficult to discern significant information from them. More clear explanation is needed, also some labels on the images would be welcome. There are no scale bars indicating the size in Fig. 23.

Response: The images in Fig. 23 has been rephotographed, as shown in Figure 4. The average folding length of the specimen at each strain rate has been measured, as shown in Table 2. It can be seen that the folding length of the specimen increases as the strain rate increases. We have added these information in the revised manuscript.

Figure 4 The specimen after dynamic compression test at various strain rates

Table 2 The folding length of the specimen at various strain rates

Strain rate(s-1)

Quasi-static

1.3×103

2.0×103

4.6×103

folding length(mm)

0.59

0.89

0.96

1.49

Comment: More discussion is needed for the explanation of the results. There is no comparison with other materials tested at high strain rates.

Response: Description and analysis are added to the "result" section. The dynamic compression properties of the honeycomb structure are related to the basis material, cell shape, cell characteristics (eg. characteristic size and cell wall thickness), processing method, impact velocity and the strain rate. It's complicated. Further studies like the influence factor of dynamic compression properties at high strain rates will be conducted in the near future.

Comment: Fig. 21. Authors observed fluctuations on stress-strain curves. They require more explanations.

Response: Figure 21 shows that there is no fluctuation at Quasi-static, only one fluctuation at strain rate 1.3×103, two fluctuations at strain rate2.0×103, and three fluctuations at strain rate 4.6×103. The fluctuation appears in the plateau stage, which is because the cellular wall instability and buckling during the compression. It can be seen that the fluctuations increases with the strain rate increases. We have added these information in the revised manuscript.

Comment: Table 5. Only average values are given. What about deviations?

Response: There is slight deviations of each specimen of the dynamic compression test at the same strain rate. The reason is because the thickness of aluminum foil varies during processing, and the boundary difference of the specimen when fabricated by wire-cutting. We have added these information in the revised manuscript.

Please see the attachment, thanks.

Reviewer 3 Report

Accepted in the present form.

Author Response

Dear  Reviewer:

Thanks very much for Accepting the manuscript.

Wish you happy new year!

Best regards.

Sheng Zhang

23/1/2020

Round 2

Reviewer 2 Report

The authors have significantly improved the article. I recommend it for publishing. 

Author Response

Dear Reviewer:

Thanks very much for accepting the manuscript. Additionally, the minor revisions of English language and style have been made and the revised parts are highlighted in the red font in the revised manuscript. 

Thanks very much and wish you happy new year!

Best regards.

Sheng Zhang

03/02/2020